# Perceptions of COVID-19 risk among individuals with preexisting health conditions

Holli A. Loomans-Kropp [1,2]*, Mohamed I. Elsaid[2,3,4,5], Jingbo Yi[2,6], Yesung Kweon[4], Electra D. Paskett[1,2], for the Impact of COVID-19 on Behaviors across the Cancer Control Continuum¶

1 Division of Cancer Prevention and Control, Department of Internal Medicine, College of Medicine, The Ohio State University Wexner Medical Center, Columbus, Ohio, United States of America,
2 Comprehensive Cancer Center, The Ohio State University, Columbus, Ohio, United States of America,
3 Division of Biostatistics and Population Health, Department of Biomedical Informatics, College of Medicine, The Ohio State University Wexner Medical Center, Columbus, Ohio, United States of America,
4 Center for Biostatistics, College of Medicine, The Ohio State University Wexner Medical Center, Columbus Ohio, United States of America, 5 Division of Medical Oncology, Department of Internal Medicine, College of Medicine, The Ohio State University Wexner Medical Center, Columbus, Ohio, United States of America, 6 Division of Epidemiology, College of Public Health, The Ohio State University, Columbus, Ohio, United States of America

¶Membership of the Impact of COVID-19 on Behaviors across the Cancer Control Continuum, full list of members is provided in the acknowledgements.
* holli.loomans-kropp@osumc.edu

## Abstract

### Objectives

To examine the association between the presence of preexisting health conditions (PEC) and the perceived risk of catching COVID-19 at the beginning of the pandemic and assess how risk perceptions changed over time.

### Methods

We used data collected as part of the "Impact of COVID-19" baseline and follow-up surveys to complete our analyses. Participants were interviewed to collect their perceptions of the risk of catching COVID-19 (baseline and follow-up) and the number and type of PEC. Kruskal-Wallis and chi-square tests were used to assess differences in baseline characteristics, and prevalence ratios were estimated using crude and adjusted modified Poisson generalized linear models.

### Results

Of the overall study population, 7,069 participants were eligible for the analysis. The majority (83.7%) of the eligible study population had a history of any PEC. Those with a history of any PEC had a median age of 58 (range: 19–97), were primarily female (67.6%), White non-Hispanic (87.8%), had some college (30.3%), were married or living as married (74.4%), lived in an urban region (67.6%), and reported

**Data availability statement:** The Impact of COVID-19 on Behaviors across the Cancer Control Continuum study is ongoing and de-identified data will become available once the study has concluded. In the interim, interested researchers can request the consortium data that support the findings by contacting the office of Recruitment, Intervention and Survey Shared Resource (RISSR) at The Ohio State University Comprehensive Cancer Center Arthur G. James Cancer Hospital and Richard J. Solove Research Institute using the procedure outlined below: 1. The researcher must submit a short proposal to the Project Publication Committee for approval. This should include the study rationale, introduction, methods, aims, data/variables, and hypothesis. 2. The Project Publication Committee will review the proposal and make suggestions and/or recommendations. 3. Once the proposal has been approved, the project statistician will start working on the analysis. The statistician and the investigators will then meet to discuss the paper and the analysis plans. The role of the statistician is to perform all analyses to ensure that methods are appropriate and statistically valid. 4. If it has been determined that the researcher will be performing the analysis (this needs to be requested in the initial proposal to the Project Publication Committee), then the researcher needs to complete the Data Distribution and Agreement Form to the Project Publication Committee. This form is a request for the specific data that the researcher needs as well as a notice of all policies and rules about using the Impact of COVID-19 data. For any data request please contact: Ms. Caroline Gault, MPH, Project Manager Recruitment, Intervention and Survey Shared Resource The Ohio State University Comprehensive Cancer Center Arthur G. James Cancer Hospital and Richard J. Solove Research Institute South Campus Gateway, 1590 N. High Street, Suite 525, Columbus, OH 43201 (614) 293-2452 caroline.gault@osumc.edu. The declared competing interests by authors do not alter our adherence to PLOS One policies on sharing data and materials.

**Funding:** This work was supported by The Ohio State University Comprehensive Cancer Center (OSUCCC) internal funding, a supplement to OSUCCC core support grant (P30 CA016058) and the OSUCCC RISSR (P30 CA016058). The Ohio State University Center for Clinical and

good (35.4%) or very good (33.9%) health. At baseline, study participants with a history of any PEC were more likely to be concerned about catching COVID-19, using a scale of 0–100, compared to those without PECs (Mean[SD] 60.8[29.8] vs. 53.2[29.7]; $p < 0.001$), as well as more likely concerned about someone they knew catching COVID-19 (Mean[SD] 70.0[28.8] vs. 64.4[29.4]; $p < 0.001$). The main effects models showed that self-concern of getting COVID-19 was higher in individuals with any PEC, compared to those with no history of PEC (Prevalence Ratio [PR], 1.15; 95%CI, 1.03–1.29); self-concern was lower at follow-up for those with any PEC, compared to baseline (PR, 0.68; 95%CI, 0.65–0.71). There was evidence of an interaction in the models of concern for self and others, suggesting that one's perception of risk was influenced by both the presence/absence of PECs and study time points.

## Conclusions

Individuals with PECs perceived a higher risk of COVID-19 infection for themselves and others towards the beginning of the pandemic, although this perception of susceptibility, or risk, was lower at follow-up. In this study, we showed that attitudes toward health and risk of disease of oneself and others may change throughout a pandemic.

## Introduction

As of early 2023, over 676 million cases of COVID-19 have been reported worldwide, a likely underreported statistic [1]. Though the outcome of COVID-19 infection is highly variable, COVID-19 infected individuals with pre-existing conditions (PECs), such as hypertension, obesity, chronic obstructive pulmonary disease, or kidney and liver disease, among other health conditions, or multimorbidity have increased risk of developing severe COVID-19 infection, poor long-term outcomes and death [2,3]. A recent comprehensive meta-analysis found that COVID-19-infected individuals with PECs had increased risk of hospitalization, intensive care unit (ICU) admission, and mortality [4]. These studies collectively suggest that PECs may contribute to COVID-19 symptom severity and risk of mortality. Over half (51.8%) of adults in the United States have at least one chronic condition and 27.2% have at least two chronic conditions [5]. A recent modeling study of prevalence data from 188 countries estimated that approximately 1.7 billion people, or 22% of the global population, has at least one chronic condition that puts them at increased risk of severe COVID-19 infection, with 4% at high risk for severe COVID-19 and hospitalization [6]. A study early in the COVID-19 pandemic found that both COVID-19-related ICU and non-ICU hospitalizations were higher among adults with PECs than those without. In the same study, the majority (94%) of deaths from COVID-19 were among individuals with at least one underlying condition [7]. An analysis of UK Biobank data found that the presence of at least one chronic condition was associated with a 1.65 times greater odds

Translational Sciences grant supports (National Center for Advancing Translational Sciences, Grant UL1TR001070) publications related to this project.The funders had no role in study design, data collection and analysis, decision to publish, or preparation of the manuscript.

**Competing interests:** Electra Paskett would like to acknowledge grants to the institution from Merck Foundation, Pfizer, Genentech, Guardant Health and Astra Zeneca and is a Member of Advisory Board for GSK and Merck. All other authors have no competing interests to declare.

(95%CI, 1.49–1.82) of severe COVID-19 infection, and that this association intensified with increasing conditions (≥4 multimorbidity index conditions adjusted OR, 2.25; 95%CI, 1.64–3.09) [8].

Despite ample research demonstrating that PECs increase susceptibility to COVID-19 infection, an individual may not perceive this increase in risk. Risk perception, or the evaluation of potential hazards, are affected by a variety of individual and societal factors, as well as can be influenced by gained knowledge, voluntary risk exposure, visibility of the risk, or trust [9,10]. This can be put in the context of the Health Belief Model, which poses that one's perceived vulnerability to disease and perceived disease severity informs the threat, informing action [11]. Of the constructs that make-up the Health Belief Model, perceived disease susceptibility and perceived disease severity contribute to risk perception, which is a significant determinant of health behaviors – all which are applicable to COVID-19 [10,12]. A recent study of individuals with chronic diseases in Central Appalachia found that actual risk of COVID-19 infection was significantly associated with higher perception of individual, but not family, friend, or community, infection risk [13]. An additional study of individuals with a history of pulmonary embolism and other PECs found that most respondents considered themselves 'high risk' for severe COVID-19 but low risk for infection [14]. These studies suggest that individual health perception may be disconnected from actual risk of COVID-19 infection. Interestingly, a community-based study in Portugal showed that poor self-reported health status was associated with a significantly increased perception of COVID-19 infection risk (OR, 2.85; 95%CI, 2.09–3.90) [15].

Furthermore, perception of COVID-19 susceptibility may impact acceptance of preventive interventions, such as vaccination. For example, a recent survey-based study found that, despite similar attitudes towards the threat of COVID-19 infection, individuals with chronic respiratory diseases were more willing to be vaccinated than those with an autoimmune disease or healthy controls, suggesting nuance in perception [16]. In the face of a pandemic, adherence to or participation in preventive measures may be impacted by perceived threat to life, as well as perception of extrinsic, or uncontrollable, risk. Perceiving a higher threat to life may increase one's practice of preventive measures, while increased perception of extrinsic risk may reduce use of these measures [17,18]. A recent systematic review found that elevated perception of COVID-19 risk led to greater use of preventive behaviors and adherence to guidelines, which, in a separate study, engagement in preventive behaviors was linked to their perceived effectiveness [19,20]. However, what remains unknown, is how behavior, and the subsequent perception of risk of infection from COVID-19, varies among individuals with and without PECs.

Our study objective was to gain additional clarity on the perception of COVID-19 infection self-risk and risk for others among individuals with PECs. We analyzed data from the "Impact of COVID-19 on Behaviors across the Cancer Control Continuum in Ohio" study, utilizing baseline and follow-up surveys to examine the association between preexisting health conditions and perceived risk of COVID-19 infection at the beginning of the pandemic, as well as how risk perceptions changed from the beginning to a later time in the pandemic.

## Materials and methods

This study was part of a National Cancer Institute-funded initiative conducted in conjunction with 16 other NCI-designated Cancer Centers – the IC-4 (Impact of COVID-19 on the Cancer Continuum Consortium). The parent and ancillary studies were originally approved by The Ohio State University (OSU) Institutional Review Board in June 2020, with approval extended annually. All participants in the study provided written or verbal informed consent. Strengthening the Reporting of Observational Studies in Epidemiology (STROBE) guidelines were followed during the completion of this study.

### Study setting and participant selection

The study design and collection have been described elsewhere [21]. The study took place in the OSU Comprehensive Cancer Center (OSUCCC) catchment area – primarily the state of Ohio with reach into adjacent areas (e.g., Indiana). Participants who agreed to be re-contacted from previous studies were asked to participate in this study, which included an array of participants. Ohio was one of the first states to issue a statewide stay-at-home order, and Ohio has many diverse populations, including the majority (50 of 88) of counties designated as rural, and 32 counties considered Appalachian, both considered medically underserved.

Eligible participants were adults aged ≥18 years who consented to participate in the study. Participants provided either written or verbal consent. Verbal consent was obtained over the phone by a trained interviewer. To ensure the inclusion of the most vulnerable, underserved, and minority populations, we sought to recruit healthy adult volunteers, cancer patients, cancer survivors, and caregivers of cancer patients and survivors in our catchment area. This was achieved by employing two recruitment strategies: 1) previous study participants in OSU studies who consented to be contacted for future research projects, and 2) our community partners and the OSUCCC Pelotonia listservs for recruitment. Study recruitment for the baseline survey began on June 19,2020, with the last completion date November 30, 2020. Those who completed the baseline were invited to complete the follow-up survey, which was open for completion from March 2, 2021, through July 7, 2021.

### Interview and data collection

Several data collection methods were used, including web, phone, and mailed surveys. Respondents with valid emails received an initial survey invitation email, with reminders sent three times, seven days apart. All participants were initially screened using an eligibility form to confirm their current Ohio or Indiana residence before conducting the survey, which could be saved and completed later. Individuals with a partial response received an email reminder one week after the last accessed date. A trained interviewer contacted participants without an email address and those with invalid emails on file by phone. Participants who were initially reached by phone were offered the option to complete the survey over the phone or online. Participants requesting mailed surveys received a cover letter and a paper survey with a self-addressed, stamped return envelope. For non-English-speaking participants, a bilingual staff member administered the survey in the appropriate language. Participants were offered a $10 gift card upon completion of the survey. All data were collected and managed using the Research Electronic Data Capture (REDCap) secure web-based application hosted at OSU.

### Study measures

To evaluate perceptions of COVID-19 risk, we used the following questions from the baseline and follow-up surveys: "From 0 to 100, how concerned are you about catching COVID-19? (0=Not at all concerned; 100=Extremely concerned)" and "From 0 to 100, how concerned are you about someone you know catching COVID-19? (0 = Not at all concerned; 100=Extremely concerned)" Response options also included 'don't know' and 'prefer not to answer'. 'High concern' about catching COVID-19 was defined as a response score of greater than or equal to 75 on a 100-point scale. This definition was applied to responses to both the baseline and follow-up surveys and for concerns both about the respondent

or someone they know catching COVID-19. The number of preexisting conditions (PECs) was assessed as continuous, dichotomous, and quartile measures. PECs included in the analysis one of the following 15 conditions: history of heart disease, high blood pressure, lung disease, diabetes, ulcer or stomach disease, kidney disease, liver disease, anemia or other blood disorder, cancer, depression, osteoarthritis or degenerative arthritis, back pain, rheumatoid arthritis, HIV, or 'other' PEC. If a participant reported at least one PEC, they were considered to have a dichotomized history of 'any comorbidity'. The number of reported PECs was also added and assessed as a continuous measure, and, from the continuous measure, divided into quartiles (cutoffs were 0–1, 2, 3, and 4+).

## Statistical analysis

Overall and PEC-stratified characteristics were summarized using descriptive statistics, with means, standard deviations, median, and range for continuous variables and proportions for categorical variables. Differences in baseline characteristics were assessed using Kruskal-Wallis and chi-square tests for continuous and categorical variables, respectively. Respondents' 'high concern' for catching COVID-19 at baseline and during follow-up was modeled as a binary outcome in logistic regression models. Multivariable models included history of any PEC as the exposure and adjusted for age, sex, race/ethnicity, marital status, insurance status, education, income, region of residence, and overall health status. Urban or rural region of residence was determined by census geocoding. Modified Poisson regression models were used to examine differences, on the prevalence ratio scale, between the history of PEC and catching COVID-19 at baseline, and follow-up, with 'high concern' modeled as a binary outcome [22]. Multivariable models also adjusted for age, sex, race/ethnicity, marital status, insurance status, education, income, region of residence, and overall health status and included exposure-time interaction terms to simultaneously assess the effect of having PECs at different time points and the perception across time for each PEC group. First-order autoregressive correlation structure was used to account for the covariance between baseline and follow-up data on the same participant. All statistical analyses were conducted using SAS v9.4, with two-tail tests and a significance level of 0.05.

## Results and discussion

From the overall study population, 7,069 participants were eligible for our analysis (Fig 1). The majority (83.7%) of the participants had a history of any PEC (Table 1). Participants with PECs were more likely to be older (Mean[SD] 58.4(12.7) vs. 47.7(12.7); $p < 0.001$), male (32.9% vs. 29.8%; $p = 0.04$), White non-Hispanic (88.5% vs. 84.3%; $p < 0.001$), less educated (13.4% vs. 8.1% high school or less; $p < 0.001$), divorced, widowed or separated (17.4% vs. 11.4%; $p < 0.001$), have both public and private insurance (29.5% vs. 10.3%; $p < 0.001$), reside in a metro area (68.86% vs. 62.9%; $p < 0.001$), have lower household income (25.5% vs. 16.3% household income < \$50,000; $p < 0.001$), and have self-reported good health (38.3% vs. 20.5%; $p < 0.001$) (Table 1). Among those with a history of at least one PEC, the most prevalent PECs were cancer (49.3%), high blood pressure (39.2%), depression (22.4%), and osteoarthritis (18.1%) (S1 Table).

We sought to examine individual perceptions of COVID-19 infection risk among individuals with and without history of PECs. Participants with a history of any PEC were more likely to be more concerned about catching COVID-19 at baseline, compared to those without a history of PECs (Mean[SD] 60.8[29.8] vs. 53.2[29.7]; $p < 0.001$), with an increased percentage with higher concern about catching COVID-19 (46.3% vs. 35.9%; $p < 0.001$) (Table 2). Similarly, those with a history of at least one PEC had higher concern about someone they knew catching COVID-19 at baseline (Mean[SD] 70.0[28.8] vs. 64.4[29.4]; $p < 0.001$), with the majority reporting high concern (60.7% vs. 51.2%; $p < 0.001$). Significantly higher self-concern was noted for all PECs except history of rheumatoid arthritis and HIV, while concern for others was associated with all except history of diabetes, rheumatoid arthritis, HIV, and liver disease (S1 Table). At follow-up, no statistical difference was observed between those with and without history of PECs for self-perceived risk (Mean[SD] 46.8[33.9] vs. 47.5[32.3]; $p = 0.70$) or perceived risk for others (Mean[SD] 61.5[32.6] vs. 62.6[32.0]; $p = 0.84$) (Fig 2). Evaluation of individual PECs showed that, despite lack of significance for the composite PEC measure for

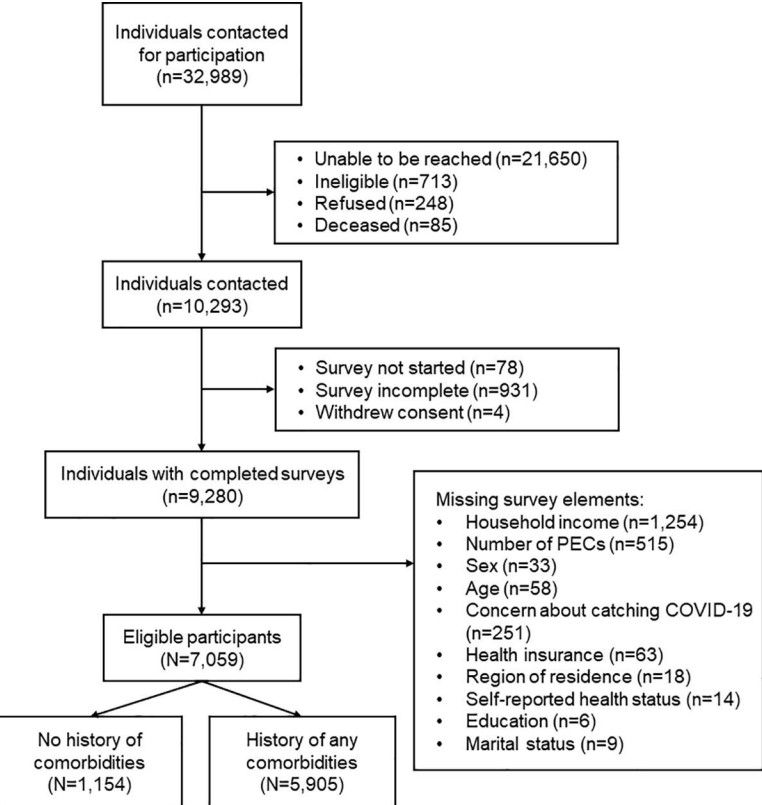

**Fig 1. Study flow chart.**

self-concern, history of lung disease (Mean[SD] 53.7[36.9] vs. 46.4[33.4]; $p = 0.003$), diabetes (Mean[SD] 50.1[34.9] vs. 46.4[33.5]; $p = 0.03$), ulcer or stomach disease (Mean[SD] 53.6[33.2] vs. 46.5[33.7]; $p = 0.006$), anemia or other blood disease (Mean[SD] 51.3[34.1] vs. 46.5[33.6]; $p = 0.02$), cancer (Mean[SD] 44.1[34.2] vs. 50.3[32.7]; $p < 0.001$), depression (Mean[SD] 49.5[33.5] vs. 46.1[33.7]; $p = 0.01$), back pain (Mean[SD] 49.3[34.7] vs. 46.1[33.3]; $p = 0.03$), rheumatoid arthritis (Mean[SD] 52.8[33.5] vs. 46.6[33.7]; $p = 0.045$), HIV (Mean[SD] 61.9[29.6] vs. 46.8[33.7]; $p = 0.049$), and other comorbidity (Mean[SD] 51.4[33.6] vs. 46.2[33.6]; $p = 0.002$) were statistically associated with higher self-perceived risk of COVID-19 at follow-up, though these PECs were highly correlated (S1 and S2 Tables). Individual history of diabetes (Mean[SD] 64.2[32.6] vs. 61.3[32.5]; $p = 0.04$), ulcer or stomach disease (Mean[SD] 71.0[31.2] vs. 61.2[32.5]; $p < 0.001$), cancer (Mean[SD] 58.8[33.2] vs. 65.1[31.4]; $p < 0.001$), depression (Mean[SD] 65.7[32.2] vs. 60.5[32.5]; $p < 0.001$), back pain (Mean[SD] 65.1[32.2] vs. 60.6[32.6]; $p < 0.001$), or other comorbidity (Mean[SD] 64.8[31.8] vs. 61.2[32.6]; $p = 0.03$) were associated with statistically increased concern for others catching COVID-19 at follow-up (S1 Table). Despite no difference history of positive COVID-19 tests at follow-up, participants with a history of PECs had moderately, though significant, higher willingness to receive a COVID-19 when one became available (93.9% vs. 90.3%; $p = 0.01$).

Using modified Poisson regression models with 'high concern' for catching COVID-19 as a binary outcome, the presence of any (≥1) PECs was not significantly associated with concern of COVID-19 infection at baseline (PR, 1.06; 95%CI, 0.95–1.19), compared to no PECs and adjusted for age, sex, race/ethnicity, marital status, insurance status, education, income, residence region, and overall health status (Table 3). Among only those with PECs, those in quartiles 3 and 4 had significantly increased concern for getting COVID-19 than those in the lower quartile (Q3: PR, 1.15; 95%CI, 1.03–1.28; Q4: PR, 1.23; 95%CI, 1.10–1.37). No significant difference was observed between quartiles 1 and 2. Multivariable logistic

**Table 1. Demographics of study participants.**

| | History of any PEC* | | Total (N = 7059) |
|---|---|---|---|
| | **No (N = 1154)** | **Yes (N = 5905)** | **Total (N = 7059)** |
| **Age, years** | | | |
| N | 1154 | 5905 | 7059 |
| Mean (SD) | 47.7 (12.70) | 58.4 (12.69) | 56.6 (13.28) |
| Median | 48.5 | 60 | 58 |
| Range | 19.0, 85.0 | 20.0, 97.0 | 19.0, 97.0 |
| **Age, years, n (%)** | | | |
| <35 | 226 (19.6%) | 301 (5.1%) | 527 (7.5%) |
| 35-49 | 379 (32.8%) | 1047 (17.7%) | 1426 (20.2%) |
| 50-59 | 331 (28.7%) | 1533 (26.0%) | 1864 (26.4%) |
| 60-69 | 176 (15.3%) | 1878 (31.8%) | 2054 (29.1%) |
| 70-79 | 39 (3.4%) | 1001 (17.0%) | 1040 (14.7%) |
| 80+ | 3 (0.3%) | 145 (2.5%) | 148 (2.1%) |
| **Sex, n (%)** | | | |
| Male | 344 (29.8%) | 1940 (32.9%) | 2284 (32.4%) |
| Female | 810 (70.2%) | 3965 (67.1%) | 4775 (67.6%) |
| **Race Ethnicity, n (%)** | | | |
| White non-Hispanic | 973 (84.3%) | 5223 (88.5%) | 6196 (87.8%) |
| Black non-Hispanic | 65 (5.6%) | 332 (5.6%) | 397 (5.6%) |
| Hispanic | 46 (4.0%) | 97 (1.6%) | 143 (2.0%) |
| Other non-Hispanic | 70 (6.1%) | 253 (4.3%) | 323 (4.6%) |
| **State, n (%)** | | | |
| Indiana | 53 (4.6%) | 206 (3.5%) | 259 (3.7%) |
| Ohio | 1101 (95.4%) | 5699 (96.5%) | 6800 (96.3%) |
| **Education, n (%)** | | | |
| HS or less | 93 (8.1%) | 789 (13.4%) | 882 (12.5%) |
| Some college/Assoc deg | 305 (26.4%) | 1832 (31.0%) | 2137 (30.3%) |
| BS/BA | 413 (35.8%) | 1683 (28.5%) | 2096 (29.7%) |
| MS/MA or more | 343 (29.7%) | 1601 (27.1%) | 1944 (27.5%) |
| **Marital status, n (%)** | | | |
| Single, never been married | 129 (11.2%) | 517 (8.8%) | 646 (9.2%) |
| Married/living as married | 894 (77.5%) | 4331 (73.8%) | 5255 (74.4%) |
| Div/wid/sep/oth | 131 (11.4%) | 1027 (17.4%) | 1158 (16.4%) |
| **Health Insurance, n (%)** | | | |
| No insurance | 84 (7.3%) | 142 (2.4%) | 226 (3.2%) |
| Public only | 115 (10.0%) | 885 (15.0%) | 1000 (14.2%) |
| Private only | 836 (72.4%) | 3135 (53.1%) | 3971 (56.3%) |
| Public and Private | 119 (10.3%) | 1743 (29.5%) | 1862 (26.4%) |
| **Region of Residence, n (%)** | | | |
| Rural | 428 (37.1%) | 1857 (31.4%) | 2285 (32.4%) |
| Urban | 726 (62.9%) | 4048 (68.6%) | 4774 (67.6%) |
| **Household Income, n (%)** | | | |
| <$35K | 79 (6.8%) | 869 (14.7%) | 948 (13.4%) |
| $35K-$49,999 | 110 (9.5%) | 640 (10.8%) | 750 (10.6%) |
| $50K-$74,999 | 213 (18.5%) | 1105 (18.7%) | 1318 (18.7%) |

*(Continued)*

**Table 1.** (Continued)

| | History of any PEC* | | |
| | No<br>(N = 1154) | Yes<br>(N = 5905) | Total<br>(N = 7059) |
|---|---|---|---|
| $75K+ | 752 (65.2%) | 3291 (55.7%) | 4043 (57.3%) |
| **Self-Reported Health Status**, n (%) | | | |
| Excellent | 311 (26.9%) | 420 (7.1%) | 731 (10.4%) |
| Very good | 551 (47.7%) | 1843 (31.2%) | 2394 (33.9%) |
| Good | 236 (20.5%) | 2260 (38.3%) | 2496 (35.4%) |
| Fair | 55 (4.8%) | 1088 (18.4%) | 1143 (16.2%) |
| Poor | 1 (0.1%) | 294 (5.0%) | 295 (4.2%) |

*PECs considered included heart disease, high blood pressure, lung disease, diabetes, ulcer of stomach disease, kidney disease, liver disease, anemia or other blood disorder, cancer, depression, osteoarthritis or degenerative arthritis, back pain, rheumatoid arthritis, HIV, or any other conditions

regression modeling showed similar results for any PEC and quartile analyses. When PECs were modeled on continuously, we found that, for each unit increase (1 PEC), concern increased 4% (PR, 1.04; 95%CI, 1.02–1.07).

Finally, we examined the interaction between PEC and time and the association with concern for catching COVID-19. The main effects models showed overall significantly increased self-concern of getting COVID-19 for individuals with any PEC (PR, 1.15; 95%CI, 1.03–1.29), compared to none, and reduced self-concern at follow-up, compared to baseline (PR, 0.68; 95%CI, 0.65–0.71) (Table 4). Concern for others contracting COVID-1D was not significant in the main effects model evaluating the association by PEC (PR, 1.07; 95%CI 0.99–1.15), though it was statistically significantly reduced at follow-up, like that observed for self-concern (PR, 0.80; 95%CI, 0.78–0.83). There was evidence of interaction between PEC and time in both the models evaluating both self-concern and concern about others.

## Conclusion

In this study, we showed that, at baseline, respondents with PECs were more concerned about catching COVID-19, compared to those without a history of PECs. Moreover, a higher number of PECs were associated with higher perceived risk, shown by comparisons across quartiles. At follow-up, however, concern for self and others was not different between those with and without PECs. These results suggest that, with the evolution of the pandemic and growing information surrounding the virus, individual risk perceptions also changed. Similar results were observed in a study conducted in Germany, where risk perception declined from March to May/June 2020 and remained low through the end of 2020 [23]. Interestingly, an analysis investigating differences in risk perception by educational status showed no difference in risk perception at baseline, but rate of decline varied between groups until stabilization.

There were many unknowns at the beginning of the COVID-19 pandemic, including routes of exposure and transmission, impact of individual risk, and long-term effects of exposure and infection. Many of these unknowns, and how they were covered in the media, may have impacted individual and group anxiety and risk perception of COVID-19 over time. In fact, an online survey of individuals in China found that repeated media exposure resulted in increased anxiety and 'vicarious trauma' [24]. In a separate study, an inverse correlation was observed between subjective well-being and COVID-19 anxiety, with an initial reduction in well-being observed at the peak of the pandemic and reduction in anxiety – thus improved well-being – occurring upon COVID-19 decline [25]. This was further shown in a study conducted in China, where, across four surveys completed between January and March 2020, perceived levels of COVID-19 infection risk and anxiety peaked at survey 2 (February 10–12), then returned to baseline or below baseline by the end of the study (March 1–10), illustrating fluctuating perceptions of COVID-19 throughout the pandemic [26].

**Table 2. Participant concern scores for catching COVID-19, stratified by history of PEC.**

| | History of PEC* | | | P-value |
|---|---|---|---|---|
| | No (N = 1154) | Yes (N = 5905) | Total (N = 7059) | |
| **Concerned about catching COVID-19 (score 0–100) (Baseline)** | | | | <.0001[1] |
| N | 1154 | 5905 | 7059 | |
| Mean (SD) | 53.2 (29.70) | 60.8 (29.84) | 59.6 (29.95) | |
| Median | 50 | 70 | 65 | |
| Range | 0.0, 100.0 | 0.0, 100.0 | 0.0, 100.0 | |
| **High concern about catching COVID-19 (Baseline), n (%)** | 414 (35.9%) | 2735 (46.3%) | 3149 (44.6%) | <.0001[2] |
| **Concerned someone you know catching COVID-19 (score 0–100) (Baseline)** | | | | <.0001[1] |
| N | 1154 | 5905 | 7059 | |
| Mean (SD) | 64.4 (29.41) | 70.0 (28.75) | 69.1 (28.94) | |
| Median | 75 | 80 | 75 | |
| Range | 0.0, 100.0 | 0.0, 100.0 | 0.0, 100.0 | |
| **High concern about someone you know catching COVID-19 (Baseline), n (%)** | 591 (51.2%) | 3585 (60.7%) | 4176 (59.2%) | <.0001[2] |
| **Concerned about catching COVID-19 (score 0–100) (follow-up)** | | | | 0.7037[1] |
| N | 426 | 3002 | 3428 | |
| Mean (SD) | 47.5 (32.27) | 46.8 (33.87) | 46.9 (33.67) | |
| Median | 50 | 50 | 50 | |
| Range | 0.0, 100.0 | 0.0, 100.0 | 0.0, 100.0 | |
| **High concern about catching COVID-19 (follow-up), n (%)** | 138 (32.4%) | 958 (31.9%) | 1096 (32.0%) | 0.8417[2] |
| **Concerned about someone you know catching COVID-19 (score 0–100) (follow-up)** | | | | 0.8391[1] |
| N | 469 | 3260 | 3729 | |
| Mean (SD) | 62.6 (31.95) | 61.5 (32.60) | 61.7 (32.52) | |
| Median | 75 | 70 | 70 | |
| Range | 0.0, 100.0 | 0.0, 100.0 | 0.0, 100.0 | |
| **High concern about someone you know catching COVID-19 (follow-up), n (%)** | 246 (52.5%) | 1603 (49.2%) | 1849 (49.6%) | 0.1840[2] |
| **Have any of your COVID-19 tests been positive? (follow-up), n (%)** | | | | 0.0727[2] |
| No | 182 (80.2%) | 1499 (84.8%) | 1681 (84.3%) | |
| Yes | 45 (19.8%) | 269 (15.2%) | 314 (15.7%) | |
| Unknown | 927 | 4137 | 5064 | |
| **Willingness to receive COVID-19 vaccine (follow-up), n (%)** | | | | 0.0134[2] |
| No | 30 (9.7%) | 178 (6.1%) | 208 (6.4%) | |
| Yes | 280 (90.3%) | 2759 (93.9%) | 3039 (93.6%) | |
| Unknown | 844 | 2968 | 3812 | |

[1]Kruskal-Wallis p-value;

[2]Chi-Square p-value.

*Comorbidities considered include: heart disease, high blood pressure, lung disease, diabetes, ulcer of stomach disease, kidney disease, liver disease, anemia or other blood disorder, cancer, depression, osteoarthritis or degenerative arthritis, back pain, rheumatoid arthritis, HIV, or any other conditions.

High concern of is derived as score 75+ to the questions "From 0 to 100, how concerned are you about catching COVID-19?" or "From 0 to 100, how concerned are you about someone you know catching COVID-19?".

Feelings of anxiety and well-being are important in one's calculation of perceived risk [27]. In the United States, among the most important predictors of risk perception are individual experience with the virus and personal knowledge of COVID-19 [28,29]. Individuals with personal connections to COVID-19 were more likely to overestimate their own likelihood or getting sick, termed the availability heuristic, or direct experience that influences one's perception of risk [30]. A

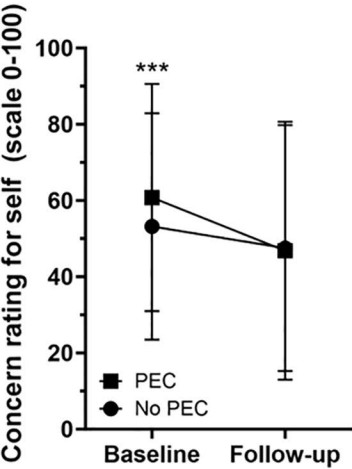
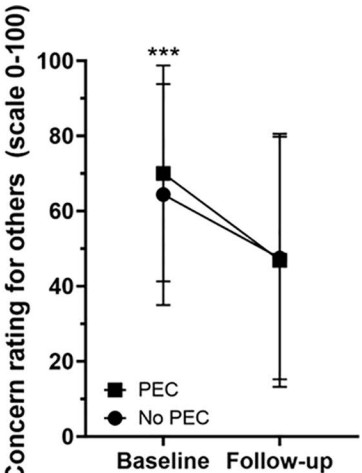

**Fig 2. Participant concern for catching COVID-19 at baseline and follow-up.** Individuals with PECs had statistically significantly increased self-concern (A) and concern for others (B) catching COVID-19 at baseline. No differences were observed at follow-up. ***$p < 0.001$.

**Table 3. Association between pre-existing health conditions and concern of getting COVID-19 at baseline.**

| PEC predictor | Crude | | | Adjusted* | | |
|---|---|---|---|---|---|---|
| | OR | 95% Lower CI | 95% Upper CI | OR | 95% Lower CI | 95% Upper CI |
| Presence of any (1+) PEC | 1.29 | 1.16 | 1.43 | 1.06 | .95 | 1.19 |
| Number of PECs, continuous | 1.09 | 1.07 | 1.11 | 1.04 | 1.02 | 1.07 |
| Number of conditions in quartile 2 (vs. quartile 1) | 1.19 | 1.08 | 1.31 | 1.09 | .99 | 1.21 |
| Number of conditions in quartile 3 (vs. quartile 1) | 1.30 | 1.18 | 1.45 | 1.15 | 1.03 | 1.28 |
| Number of conditions in quartile 4 (vs. quartile 1) | 1.50 | 1.37 | 1.64 | 1.23 | 1.10 | 1.37 |

PEC, pre-existing condition; OR, odds ratio.

The binary outcome is modeled as high concern of catching COVID, derived as score 75+ on a 100-point scale. PR is modeled using a Poisson generalized linear mode.

*Adjusted for age, sex, race/ethnicity, marital status, insurance status, education, income, region of residence, and overall health status.

university-based study found increased perceived risk among students with high levels of information-seeking behaviors, however information seeking more highly influenced collective risk perceptions, rather than individual, and more so among individuals with high social media use [29]. Similar results were observed among adolescents and young adults, where elevated risk perception was associated with higher disease knowledge, and higher risk levels were perceived for others, rather than for self [31].

However, we were interested in better understanding the influence of perceived risk among individuals with PECs, as over half of adults have a chronic condition [5]. We found that perceived COVID-19 infection self-risk and infection risk for others was higher among individuals with PECs at baseline, compared to those without a history of PECs. These data are consistent with other studies. For example, a questionnaire-based study of individuals living in Central Appalachia found a significant positive association between actual risk – based on the number of experienced chronic conditions – and perceptions of self-risk [13]. Similarly, a study based in Portugal found that perception of risk of severe COVID-19 was higher for older adults with at least one chronic condition, compared to those without [15]. Interestingly, individuals with PECs and those with a poor health status were also at increased risk of depression, anxiety, fear of COVID-19, and dangerous

**Table 4. Association of PEC and concern of catching COVID-19 across time.**

| | Main effects* | | | PEC x time interaction** | | | | | |
|---|---|---|---|---|---|---|---|---|---|
| | PR | 95% Lower CI | 95% Upper CI | PR | 95% Lower CI | 95% Upper CI | PR | 95% Lower CI | 95% Upper CI |
| **Concern about self** | | | | | | | | | |
| PEC | *All* | | | Baseline | | | Follow-Up | | |
| Any vs. none | 1.15 | 1.03 | 1.29 | 1.27 | 1.11 | 1.44 | 0.97 | 0.84 | 1.13 |
| Time | *All* | | | No PEC | | | Any PEC | | |
| Follow-up vs. Baseline | 0.68 | 0.65 | 0.71 | 0.86 | 0.74 | 0.99 | 0.66 | 0.62 | 0.69 |
| **Concern about others** | | | | | | | | | |
| PEC | *All* | | | Baseline | | | Follow-Up | | |
| Any vs. none | 1.07 | 0.99 | 1.15 | 1.14 | 1.04 | 1.24 | 0.98 | 0.89 | 1.08 |
| Time | *All* | | | No PEC | | | Any PEC | | |
| Follow-up vs. Baseline | 0.80 | 0.78 | 0.83 | 0.91 | 0.83 | 1.00 | 0.79 | 0.76 | 0.81 |

PEC, pre-existing condition; PR, prevalence ratio.

The binary outcome is modeled as high concern of catching COVID, derived as score 75+on a 100-point scale. PR is modeled using a Poisson generalized linear model.

*Adjusted for concern at baseline vs. follow-up time points, age, sex, race/ethnicity, marital status, insurance status, education, income, region of residence, and overall health status.

**Adjusted for concern at baseline vs. follow-up time points and its interaction with PEC status, age, sex, race/ethnicity, marital status, insurance status, education, income, region of residence, and overall health status.

alcohol use [15]. Lastly, a qualitative analysis suggested that, with substantial changing information, participants would choose guidance based on their own perceived risk [32]. The study also found differences in perceptions by age – older individuals thought they were less likely to be infected than younger people, despite an increased risk of infection, and considered themselves to be in good health [32].

We also found that, at follow-up, perceived risk of COVID-19 infection was no different between individuals with and without a history of PECs. This may be due to increasing knowledge and information accessibility as the pandemic progressed, which may have been fed by exposure to the media and other new sources. For instance, high consumption of media and other sources of information influenced risk perception in a cross-sectional study of pharmacists [33]. An additional study found that educational intervention (e.g., video on COVID-19 etiology, transmission, symptoms, etc.) could improve self-resilience, perception, and safety [34]. With continuing coverage of COVID-19 over the last four years, an abundance of information has become available to the public, thus further influencing the perception of risk over time. Moreover, the COVID-19 vaccine rollout in Ohio was initiated in December 2020, after the baseline survey, and may have influenced perceptions of COVID-19 risk reported in the follow-up survey, as, by March 2021, many of the individuals included in our study may have been eligible for vaccination. Higher perceived likelihood of COVID-19 infection, perceived severity of COVID-19 infection, consequences of infection, or seeing COVID-19 as a 'threat' were associated with higher vaccine acceptance and intention [35–37]. In parallel, Sepucha et al. found that COVID-19 risk perceptions, as well as uncertainty and anxiety, were relatively stable across surveys completed in 2020, but showed declining perception of risk in 2021, which coincided with the vaccination rollout [38]. Following vaccine rollout, a national survey of adults reported significant increases in engagement in social activities, such as indoor dining, socializing with friends, and grocery store visits, though vaccinated individuals participated in these activities less than unvaccinated individuals [39]. Collectively, these data suggest that the evolution of the COVID-19 pandemic, along with increased information and vaccine availability, may influence one's perception of risk of COVID-19 infection.

Though we were able to gain substantial insight into COVID-19 risk perception, there are limitations in this study to note. PECs were self-reported by participants and might be subject to issues with recall or underreporting. This may be

partially reflected in the characteristics of our study population, as the prevalence of chronic conditions in the cohort was notably higher than reported in the U.S. population, though this is likely due to incorporating a greater number of PECs in our analysis, as well as slight differences in the types of PECs reported [5]. Though this difference is notable, we believe that including more PECs, and PECs that occur frequently, may allow for better generalizability to the wider U.S. population. Furthermore, there is a possibility of selection (ascertainment) or sampling bias, as individuals who participated in the study were participants in prior studies at OSUCCC, thus missing individuals who sought care elsewhere, and potentially limiting generalizability. Lastly, it is possible that seasonality influenced perception of COVID-19 infection risk, as prior studies in other infectious diseases, such as seasonal influenza, have demonstrated that risk perception is dynamic [40,41]. Similarly, COVID-19 exhibits seasonal differences in infection rates, with spikes in cases observed between November and January [42]. A strength of our study was the collection of two time points, providing the opportunity to explore changes in perceptions over time amid a pandemic. Additionally, the study structure and catchment area allowed us to gain insight into risk perceptions of individuals who identify as underrepresented or medically underserved, populations typically not captured in similar studies. Approximately 12% of our study population did not identify as White non-Hispanic and 32% resided in rural areas, statistics that are reflective of the Ohio population (Table 1). Finally, we were able to evaluate perceptions of risk by individual PEC, also suggesting that the severity of the experienced PEC may influence individual risk perception.

In conclusion, individuals with PECs perceived an increased risk of COVID-19 infection for themselves and others early in the pandemic. At follow-up, perceived risk of COVID-19 infection was not different between those with or without PECs. We demonstrate the changing attitudes of health and disease risk throughout a pandemic. The results of this study may help inform future health-related communication strategies, as well as targeted messaging for those with PECs, in response to disease.

## Supporting information

**S1 Table. Participant concern scores by individual PEC.**
(XLSX)

**S2 Table. PEC correlation matrix.**
(XLSX)

**S1 Checklist. STROBE checklist pub.**
(DOCX)

## Acknowledgments

We would like to thank those who helped and supported the Impact of COVID-19 on Behaviors across the Cancer Control Continuum in Ohio and Indiana Consortium, including Cecilia DeGraffinreid and Chastity Washington.

## Author contributions

**Conceptualization:** Holli A. Loomans-Kropp, Mohamed I. Elsaid.

**Formal analysis:** Mohamed I. Elsaid, Jingbo Yi, Yesung Kweon.

**Funding acquisition:** Electra D. Paskett.

**Investigation:** Holli A. Loomans-Kropp.

**Methodology:** Holli A. Loomans-Kropp, Mohamed I. Elsaid.

**Supervision:** Holli A. Loomans-Kropp, Mohamed I. Elsaid, Electra D. Paskett.

**Writing – original draft:** Holli A. Loomans-Kropp, Jingbo Yi.

**Writing – review & editing:** Holli A. Loomans-Kropp, Mohamed I. Elsaid, Yesung Kweon, Electra D. Paskett.

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
