## [Decision Letter · Decision Letter 0]

8 Dec 2024

PONE-D-24-38967Perceptions of COVID-19 risk among individual with preexisting health conditionsPLOS ONE

Dear Dr. Loomans-Kropp,

Thank you for submitting your manuscript to PLOS ONE. After careful consideration, we feel that it has merit but does not fully meet PLOS ONE’s publication criteria as it currently stands. Therefore, we invite you to submit a revised version of the manuscript that addresses the points raised during the review process.

Dear authors,Many thanks for submitting your manuscript to PLOS ONE. The manuscript has now been reviewed by two expert reviewers.As you will from their comments below, while they both appraciated your work, they also raised a number of concerns that need to be addressed before the manuscript can be considered for publication. I look forward to receiving a revised version of the manuscript. Best wishesNicola Diviani==============================

We look forward to receiving your revised manuscript.

Kind regards,

Nicola Diviani

Academic Editor

PLOS ONE

Journal Requirements:

2. Thank you for stating the following financial disclosure: This work was supported by The Ohio State University Comprehensive Cancer Center (OSUCCC) internal funding, a supplement to OSUCCC core support grant (P30 CA016058) and the OSUCCC RISSR (P30 CA016058). The Ohio State University Center for Clinical and Translational Sciences grant supports (National Center for Advancing Translational Sciences, Grant UL1TR001070) publications related to this project.  

3. Thank you for stating the following in the Competing Interests section: Electra Paskett would like to acknowledge grants to the institution from Merck Foundation, Pfizer, Genentech, Guardant Health and Astra Zeneca and is a Member of Advisory Board for GSK and Merck. All other authors have no competing interests to declare.

Reviewers' comments:

Reviewer's Responses to Questions

**Comments to the Author**

1. Is the manuscript technically sound, and do the data support the conclusions?

Reviewer #1: Yes

Reviewer #2: Partly

2. Has the statistical analysis been performed appropriately and rigorously? 

Reviewer #1: Yes

Reviewer #2: Yes

3. Have the authors made all data underlying the findings in their manuscript fully available?

Reviewer #1: No

Reviewer #2: No

4. Is the manuscript presented in an intelligible fashion and written in standard English?

Reviewer #1: Yes

Reviewer #2: Yes

5. Review Comments to the Author

Reviewer #1: Review

PLOS ONE

Perceptions of COVID-19 risk among individual with preexisting health conditions

This manuscript investigates the relationship between preexisting health conditions (PECs) and perceptions of COVID-19-related risk in the early stages of the global pandemic, as well as how these perceptions changed over time. By analysing data from baseline and follow-up surveys, the authors explore differences in risk perceptions among individuals with and without PECs. The paper considers an important issue and addresses its research questions sensibly. It employs a good sample size and contributes meaningfully to the field of risk perceptions research. However, it needs to address the following points before publication is considered. In particular, greater clarity in defining the category of risk perception the manuscript is addressing, and greater justification of the analytical strategy would add to the manuscript. I hope the following suggestions can contribute to an enhanced manuscript.

Abstract

Some more clarity earlier on in the abstract would help “perceptions of the risk of COVID-19” are not clarified until the results in the abstract, where you state “catching COVID-19”. The literature around risk perception during COVID is extensive, therefore it is important to specify the ‘perception of what’ you are addressing. Precision of what you are measuring in the method would add clarity, as would information about the size and nature of your sample.

“The main effects models showed overall significantly increased self-concern of getting COVID-19 for individuals with any PEC (PR, 1.15; 95%CI, 1.03- 1.29), compared to none, and reduced self-concern at follow-up, compared to baseline (PR, 0.68; 95%CI, 0.65-0.71).”

This is a confusing sentence. I think the confusion comes from the use of ‘increased’ which would suggest change over time, but then you highlight there is overall reduction from baseline to follow-up, which suggests you are initially referring to a difference between groups. I suggest “self-concern of getting COVID-19 is higher in individuals with any PEC, compared to those with none” to avoid potential confusion.

Introduction

Is it right that “the presence of pre-existing conditions may exacerbate infection” or is it that it may exacerbate the extent of the potential detrimental effects of the infection? This is a point that needs greater clarity throughout, I would suggest stressing that the concern around people with PECs is that they are more likely to experience severe symptoms from COVID-19 than those without PECs. Also, in the introduction you are sometimes referring to risk of infection, risk of severe symptoms, and risk of fatality, so avoid using “perception of COVID-19 risk” alone, to increase clarity.

You state “Over half (51.8%) of adults” … is this globally, or in a specific country?

You mention that “perception of risk may impact acceptance of preventive interventions” an important follow-up might be to mention that though perceived risk of contracting COVID-19 may increase adherence to preventative hygiene behaviours (mask wearing etc), it may lead to a reduction in health effort and poorer health behaviours due to a decreased sense of control. See the body of literature around this topic for suggested citations:

COVID-19: the relationship between perceptions of risk and behaviours during lockdown

https://doi.org/10.1007/s10389-021-01543-9

The Relationship Between Perceived Uncontrollable Mortality Risk and Health Effort: Replication, Secondary Analysis, and Mini Meta-analysis

https://doi.org/10.1093/abm/kaad072

The Uncontrollable Mortality Risk Hypothesis: Theoretical foundations and implications for public health

https://doi.org/10.1093/emph/eoae009

Method

You state that you “examine the association between 85 preexisting health conditions and perceived risk of COVID-19”. I think the use of ‘association’ is unclear/misleading, in the context of statistical associations, as you are mostly looking at differences between those with and without PECs, not strictly the association between variables.

“Differences in baseline characteristics were assessed using Kruskal-Wallis and chi-square tests for continuous and categorical variables, respectively” This is fine, but perhaps inform the reader why Kruskal-Wallis – were ANOVA assumptions not met?

“Respondents’ ‘high concern’ for catching COVID-19 at baseline and during follow-up was modelled as a binary outcome in logistic regression models.” I think the reader would be interested to learn your rationale for not treating perceived risk of catching COVID simply as a continuous variable, what is gained by treating it as a binary variable, what is lost, and why the threshold of 75 (well at least I’d be interested to know)?

Results

“(83.7%) of the participants had a history of any PEC” - This is high. Not necessarily a problem, but different to the statistics you cite in your introduction. Presumably as a result of your sampling? A sentence to acknowledge/discuss the impact that this may have on your findings or their generalisability would be useful.

For the evaluation of the individual PECs. It is not very informative to only provide the p values, without the effect size, please include this for greater statistical transparency.

Table 1 - The inclusion of p values is confusing without having to jump up and down to see what test you are referring to, and with no test statistic. I suggest simply reporting the descriptive sample characteristics, then moving on to inferential models.

Discussion

To what extent might seasonal effects have impacted the overall decline in concern between baseline and follow-up? As people are more susceptible to respiratory infections in the winter months, the baseline data collection (June 19, 2020 to November 30, 2020), leading into winter may, have resulted in a higher level of risk perception, compared to the follow-up in march to July.

More likely, and more important to discuss, what impact do you think the vaccine rollout had on risk perceptions during this time. I don’t know about the availability of the vaccine in Ohio at the time of data collection, but I imagine that people with PECs were likely to have been prioritised over people without PECs. Perhaps people with PECs were more likely to have received a vaccine than those without, potentially levelling out the risk between groups. This potentially altered the actual risk of contracting COVID-19 between those with/without PECs.

These are not criticisms of the study, but points that warrant some acknowledgement/discussion.

A final point regarding your discussion of the wider topic of risk perception… when discussing the overall decline in perceived risk of contracting COVID-19, it would be sensible to highlight literature pointing out this as a normal feature in risk perception – that novel risks are perceived as more threatening when they first appear, but often decline as prevalence increases and the population becomes more familiar with that risk.

This leads nicely to discussions around the primary bias in risk perception, which would be pertinent here:

Judged frequency of lethal events

https://doi.org/10.1037/0278-7393.4.6.551

Perceptions of control over different causes of death and the accuracy of risk estimations

https://link.springer.com/article/10.1007/s10389-023-01910-8

PLOS ONE Criteria:

1. The study presents the results of original research. YES

2. Results reported have not been published elsewhere. AS FAR AS I AM AWARE

3. Experiments, statistics, and other analyses are performed to a high technical standard and are described in sufficient detail. YES – WITH SOME COMMENTS TO BE ADDRESSED

4. Conclusions are presented in an appropriate fashion and are supported by the data. YES – WITH SOME COMMENTS TO BE ADDRESSED

5. The article is presented in an intelligible fashion and is written in standard English. YES

6. The research meets all applicable standards for the ethics of experimentation and research integrity. AS FAR AS I AM AWARE

7. The article adheres to appropriate reporting guidelines and community standards for data availability. AS FAR AS I AM AWARE

Reviewer #2: Thank you for the opportunity to review this paper, which investigates risk perception during the COVID-19 pandemic with a specific focus on preexisting health conditions (PEC). I believe this is a timely and relevant topic, as understanding individual and social phenomena during the pandemic is crucial for future pandemic preparedness. The study's large sample size and longitudinal design are commendable strengths, allowing for a robust exploration of the research question. However, there are several issues, both major and minor, that need to be addressed to improve the clarity, rigor, and overall contribution of the paper.

Abstract

• The conclusion in the abstract emphasizes the role of experience and increased knowledge in shaping risk perception. While this interpretation is interesting, it appears to go beyond what is directly supported by the study's findings. I recommend revising this to more closely align with the results.

Introduction

• Theoretical Framework: The introduction would benefit from referencing established theoretical frameworks related to risk perception. Specifically:

o When discussing the self-perceived risk of contracting COVID-19, it would be appropriate to use the term "susceptibility".

o Similarly, the self-perceived risk of fatality should be referred to as "severity".

o These terms align with common frameworks such as the Health Belief Model and provide a more precise context for the discussion.

• Existing Literature: While the study aims to contribute to the understanding of risk perception during pandemics, the introduction could better situate this research within the existing body of evidence. Incorporating references to previous studies on pandemic-related risk perception would help highlight the study's unique contributions.

Results

• Results should be presented objectively, without interpretation or commentary. For example, in line 75, the use of the word "interestingly" introduces a subjective tone that is best reserved for the discussion section. I recommend revising such language to maintain a neutral reporting style.

Discussion

• Sample Characteristics: The paper acknowledges the large sample size as a strength, but it is notable that many respondents reported a history of PEC. While self-reporting is identified as a potential limitation, this point requires further discussion. For instance:

o Could the high prevalence of PEC in the sample introduce bias or affect generalizability?

o What are the potential implications for interpreting the findings?

• Additional Limitations: While the limitations section touches on underreporting, there are likely other limitations that merit discussion, such as possible sampling biases or challenges inherent in longitudinal designs (e.g., attrition).

• Knowledge and Risk Perception: The interpretation linking changes in risk perception to increased knowledge requires further justification.

o Why is this claim made, and how is it supported by the data?

o To strengthen this interpretation, it would be helpful to introduce the construct of "knowledge about pandemics" in the introduction and discuss its potential impact on risk perception.

Conclusion

• I would suggest separating and titling the conclusion section. Ensure that the conclusions remain closely tied to the study’s findings and avoid overgeneralizing. It is important to clearly articulate what is supported by the data and what remains speculative.

6. PLOS authors have the option to publish the peer review history of their article (what does this mean? ). If published, this will include your full peer review and any attached files.

**Do you want your identity to be public for this peer review?** For information about this choice, including consent withdrawal, please see our Privacy Policy .

Reviewer #1: **Yes: ** Dr Richard Brown

Reviewer #2: No

---

## [Author Response · Author response to Decision Letter 0]

24 Jan 2025

Editorial comments:

Response: We have updated the manuscript format and file names to adhere to PLOS ONE guidelines.

2. Thank you for stating the following financial disclosure: “This work was supported by The Ohio State University Comprehensive Cancer Center (OSUCCC) internal funding, a supplement to OSUCCC core support grant (P30 CA016058) and the OSUCCC RISSR (P30 CA016058). The Ohio State University Center for Clinical and Translational Sciences grant supports (National Center for Advancing Translational Sciences, Grant UL1TR001070) publications related to this project.” Please state what role the funders took in the study. If the funders had no role, please state: ""The funders had no role in study design, data collection and analysis, decision to publish, or preparation of the manuscript."" If this statement is not correct you must amend it as needed. Please include this amended Role of Funder statement in your cover letter; we will change the online submission form on your behalf.

Response: Thank you, we have added the Role of the Funder statement in the revised cover letter.

3. Thank you for stating the following in the Competing Interests section: “Electra Paskett would like to acknowledge grants to the institution from Merck Foundation, Pfizer, Genentech, Guardant Health and Astra Zeneca and is a Member of Advisory Board for GSK and Merck. All other authors have no competing interests to declare.” Please confirm that this does not alter your adherence to all PLOS ONE policies on sharing data and materials, by including the following statement: ""This does not alter our adherence to PLOS ONE policies on sharing data and materials.” (as detailed online in our guide for authors http://journals.plos.org/plosone/s/competing-interests). If there are restrictions on sharing of data and/or materials, please state these. Please note that we cannot proceed with consideration of your article until this information has been declared. Please include your updated Competing Interests statement in your cover letter; we will change the online submission form on your behalf.

Response: We have added the Competing Interests statement in the cover letter and in the Data Availability statement in the manuscript.

Response: The Impact of COVID-19 on Behaviors across the Cancer Control Continuum Study is ongoing and, thus, the data has not been released or made publicly available. However, the data is available upon request using the procedure outlined in the manuscript in the Data Availability Statement. This procedure is similar to that reported in Habtemariam et al. 2024 (PMID: 38820378).

Response: We have added the full name, date of IRB approval, and language on participant consent in the Methods section (lines 107-109).

Reviewer 1 comments:

1. Abstract - Some more clarity earlier on in the abstract would help “perceptions of the risk of COVID-19” are not clarified until the results in the abstract, where you state “catching COVID-19”. The literature around risk perception during COVID is extensive, therefore it is important to specify the ‘perception of what’ you are addressing. Precision of what you are measuring in the method would add clarity, as would information about the size and nature of your sample.

Response: We thank the reviewer for the insightful comment. To address this, we have edited the language in the Abstract to clarify that by ‘perceived risk of COVID-19,’ we mean susceptibility to or catching COVID-19. We have also added additional information regarding our study population and edited the language throughout the manuscript to provide additional clarity.

2. Abstract - “The main effects models showed overall significantly increased self-concern of getting COVID-19 for individuals with any PEC (PR, 1.15; 95%CI, 1.03- 1.29), compared to none, and reduced self-concern at follow-up, compared to baseline (PR, 0.68; 95%CI, 0.65-0.71).” This is a confusing sentence. I think the confusion comes from the use of ‘increased’ which would suggest change over time, but then you highlight there is overall reduction from baseline to follow-up, which suggests you are initially referring to a difference between groups. I suggest “self-concern of getting COVID-19 is higher in individuals with any PEC, compared to those with none” to avoid potential confusion.

Response: We appreciate the comment and have edited this sentence to improve clarity and readability.

3. Introduction - Is it right that “the presence of pre-existing conditions may exacerbate infection” or is it that it may exacerbate the extent of the potential detrimental effects of the infection? This is a point that needs greater clarity throughout, I would suggest stressing that the concern around people with PECs is that they are more likely to experience severe symptoms from COVID-19 than those without PECs. Also, in the introduction you are sometimes referring to risk of infection, risk of severe symptoms, and risk of fatality, so avoid using “perception of COVID-19 risk” alone, to increase clarity.

Response: We thank the reviewer for this important comment, and we have clarified these points in the introduction. Several studies have demonstrated that COVID-19 infection often results in increased severity of infection and worse patient outcomes in those with PECs or multimorbidity than those without PECs (Chatterjee et al. 2023 ACS Pharmacol Transl Sci; Thierry et al. 2020 Precis Clin Med; Treskova-Schwarzbach et al. 2021 BMC Med). Additional studies that support this assertion have been included and cited in lines 55-61.

4. Introduction - You state “Over half (51.8%) of adults” … is this globally, or in a specific country?

Response: This statistic is from an analysis of the 2018 United States National Health Interview Survey (NHIS). Specifically, in this analysis, the authors considered 10 specific chronic conditions, including arthritis, cancer, chronic obstructive pulmonary disease, and diabetes, among others. Additional studies have supported this finding, showing, for example, in the Medical Expenditure Panel Survey, that 52.6% of participants had at least one chronic condition (Schlitz 2022, Front Public Health). We have added “in the United States” to the Introduction on line 61.

5. Introduction - You mention that “perception of risk may impact acceptance of preventive interventions” an important follow-up might be to mention that though perceived risk of contracting COVID-19 may increase adherence to preventative hygiene behaviours (mask wearing etc), it may lead to a reduction in health effort and poorer health behaviours due to a decreased sense of control. See the body of literature around this topic for suggested citations: COVID-19: the relationship between perceptions of risk and behaviours during lockdown (https://doi.org/10.1007/s10389-021-01543-9); The Relationship Between Perceived Uncontrollable Mortality Risk and Health Effort: Replication, Secondary Analysis, and Mini Meta-analysis

(https://doi.org/10.1093/abm/kaad072); The Uncontrollable Mortality Risk Hypothesis: Theoretical foundations and implications for public health (https://doi.org/10.1093/emph/eoae009).

Response: We thank the reviewer for the comment and have included additional information and references to this effect in the Introduction (lines 89-95).

6. Methods - You state that you “examine the association between 85 preexisting health conditions and perceived risk of COVID-19”. I think the use of ‘association’ is unclear/misleading, in the context of statistical associations, as you are mostly looking at differences between those with and without PECs, not strictly the association between variables.

Response: We have edited the language in the Methods to “examine differences… in history of PEC and catching COVID-19” (line 166).

7. Methods - “Differences in baseline characteristics were assessed using Kruskal-Wallis and chi-square tests for continuous and categorical variables, respectively” This is fine, but perhaps inform the reader why Kruskal-Wallis – were ANOVA assumptions not met?

Response: Due to the large number of variables examined, we opted to use nonparametric tests, such as the Kruskal-Wallis test, as the ANOVA assumptions for several variables were not met.

8. Methods - “Respondents’ ‘high concern’ for catching COVID-19 at baseline and during follow-up was modelled as a binary outcome in logistic regression models.” I think the reader would be interested to learn your rationale for not treating perceived risk of catching COVID simply as a continuous variable, what is gained by treating it as a binary variable, what is lost, and why the threshold of 75 (well at least I’d be interested to know)?

Response: We provided ‘concern’ as a continuous variable in sensitivity analyses (see Table 2) as well as in our initial regression models, shown in Table 3. The binary variable of ‘concern’ was chosen for interpretability, and we also wanted to infer the predictors for having overall “high” concern, rather than per incremental increase in concern level.

9. Results - “(83.7%) of the participants had a history of any PEC” - This is high. Not necessarily a problem, but different to the statistics you cite in your introduction. Presumably as a result of your sampling? A sentence to acknowledge/discuss the impact that this may have on your findings or their generalisability would be useful.

Response: In our study, we considered 15 different chronic conditions in our analysis (including an ‘other PEC’ category, which included other conditions not listed), compared to only 10 chronic conditions considered in Boersma et al. 2020. The differences in our categorization of PEC, compared to Boersma et al., are likely to account for the differences in PEC prevalence in our study population and that reported in the U.S. population. To clarify this point, we have included additional details on the PECs in the Discussion (lines 310-315).

10. Results - For the evaluation of the individual PECs. It is not very informative to only provide the p values, without the effect size, please include this for greater statistical transparency.

Response: We thank the reviewer for the comment and have included the requested effect sizes in the Results (lines 197-211).

11. Table 1 - The inclusion of p values is confusing without having to jump up and down to see what test you are referring to, and with no test statistic. I suggest simply reporting the descriptive sample characteristics, then moving on to inferential models.

Response: We have removed the p-values from Table 1.

12. Discussion - To what extent might seasonal effects have impacted the overall decline in concern between baseline and follow-up? As people are more susceptible to respiratory infections in the winter months, the baseline data collection (June 19, 2020, to November 30, 2020), leading into winter may, have resulted in a higher level of risk perception, compared to the follow-up in March to July. More likely, and more important to discuss, what impact do you think the vaccine rollout had on risk perceptions during this time. I don’t know about the availability of the vaccine in Ohio at the time of data collection, but I imagine that people with PECs were likely to have been prioritised over people without PECs. Perhaps people with PECs were more likely to have received a vaccine than those without, potentially levelling out the risk between groups. This potentially altered the actual risk of contracting COVID-19 between those with/without PECs. These are not criticisms of the study, but points that warrant some acknowledgement/discussion.

Response: We thank the reviewer for the comment and believe it raises an important discussion point. As suggested by the reviewer, seasonality may have an impact on one’s perceived risk of infection. A study investigating seasonal influenza and cold risk found that risk perception was dynamic over time, with lower perception risk (infection) in the 2006 and 2016 waves and higher risk perception in the 2009 and 2018 waves (Lages et al. 2021 Risk Analysis). Like other infectious diseases, such as influenza and the cold, COVID-19 exhibits seasonal differences in infection rates, with spikes in cases observed from November to January (Wiemken et al. 2023 Scientific Reports). Therefore, it stands to reason that perception of COVID-19 infection risk may not have only been influenced by the novelty of the virus, but by prior experience with seasonal infections. We have added text in the discussion to illustrate this point (lines 318-322).

Regarding the impact of vaccine availability on COVID-19 risk perception, we agree that it is possible that the availability of and access to the vaccine may have influenced perceptions at follow-up. Initial vaccine rollout occurred in Ohio in mid-December 2020, after the completion of the baseline survey but before the distribution of the follow-up survey. Several studies have shown that individuals who perceived a higher likelihood of contracting COVID-19, severity of the disease, and higher anxiety surrounding COVID-19 had higher vaccine acceptance, intention, and uptake (Hilverda & Vollmann 2021 Vaccines; Fleury-Bahi et al. 2023 Front Psychol; Reiter et al. 2020 Vaccine). In our study, if participants had been vaccinated, or had access to the vaccination, their perception of COVID-19 risk may have changed since the beginning of the pandemic. Moreover, receiving the vaccine may have led to participation in behaviors post-vaccination that would previously been perceived as ‘risky’. We have discussed the possibility of vaccine availability and access in the Discussion (lines 295-301).

13. Discus

---

## [Decision Letter · Decision Letter 1]

25 Feb 2025

Perceptions of COVID-19 risk among individuals with preexisting health conditions

PONE-D-24-38967R1

Dear Dr. Loomans,

We’re pleased to inform you that your manuscript has been judged scientifically suitable for publication and will be formally accepted for publication once it meets all outstanding technical requirements.

Kind regards,

Nicola Diviani

Academic Editor

PLOS ONE

Additional Editor Comments (optional):

Reviewers' comments:

Reviewer's Responses to Questions

**Comments to the Author**

1. If the authors have adequately addressed your comments raised in a previous round of review and you feel that this manuscript is now acceptable for publication, you may indicate that here to bypass the “Comments to the Author” section, enter your conflict of interest statement in the “Confidential to Editor” section, and submit your "Accept" recommendation.

Reviewer #1: All comments have been addressed

Reviewer #2: All comments have been addressed

2. Is the manuscript technically sound, and do the data support the conclusions?

Reviewer #1: Yes

Reviewer #2: Yes

3. Has the statistical analysis been performed appropriately and rigorously? 

Reviewer #1: Yes

Reviewer #2: Yes

4. Have the authors made all data underlying the findings in their manuscript fully available?

Reviewer #1: (No Response)

Reviewer #2: No

5. Is the manuscript presented in an intelligible fashion and written in standard English?

Reviewer #1: Yes

Reviewer #2: Yes

6. Review Comments to the Author

Reviewer #1: The authors have done a wonderful job at attending to all comments with a comprehensive and sensible approach. This provides a valuable contribution to the literature.

Reviewer #2: The authors addressed all comments of my previous review and the ones of the other reviewers. The manuscript has improved, especially the introduction and discussion sections.

7. PLOS authors have the option to publish the peer review history of their article (what does this mean? ). If published, this will include your full peer review and any attached files.

**Do you want your identity to be public for this peer review?** For information about this choice, including consent withdrawal, please see our Privacy Policy .

Reviewer #1: **Yes: ** Richard Brown

Reviewer #2: No

---

## [Editor Report · Acceptance letter]

PONE-D-24-38967R1

PLOS ONE

Dear Dr. Loomans-Kropp,

I'm pleased to inform you that your manuscript has been deemed suitable for publication in PLOS ONE. Congratulations! Your manuscript is now being handed over to our production team.

Kind regards,

on behalf of

Dr. Nicola Diviani

Academic Editor

PLOS ONE